# Sulfonated Azocalix[4]arene-Modified Metal–Organic Framework Nanosheets for Doxorubicin Removal from Serum

**DOI:** 10.3390/nano14100864

**Published:** 2024-05-16

**Authors:** Xiao-Min Cao, Yuan-Qiu Cheng, Meng-Meng Chen, Shun-Yu Yao, An-Kang Ying, Xiu-Zhen Wang, Dong-Sheng Guo, Yue Li

**Affiliations:** 1College of Chemistry, Nankai University, Tianjin 300071, China; 2120210908@mail.nankai.edu.cn (X.-M.C.); yqcheng@mail.nankai.edu.cn (Y.-Q.C.); 1120200366@mail.nankai.edu.cn (M.-M.C.); yaosy@mail.nankai.edu.cn (S.-Y.Y.); yak147369@163.com (A.-K.Y.); 2120221095@mail.nankai.edu.cn (X.-Z.W.); 2State Key Laboratory of Elemento-Organic Chemistry, Key Laboratory of Functional Polymer Materials (Ministry of Education), Frontiers Science Center for New Organic Matter, Collaborative Innovation Center of Chemical Science and Engineering (Tianjin), Nankai University, Tianjin 300071, China; 3Xinjiang Key Laboratory of Novel Functional Materials Chemistry, College of Chemistry, and Environmental Sciences, Kashi University, Kashi 844000, China; 4Key Laboratory of Advanced Energy Materials Chemistry (Ministry of Education), Nankai University, Tianjin 300071, China

**Keywords:** metal–organic framework, nanosheet, sulfonated azocalix[4]arene, adsorption, doxorubicin

## Abstract

Chemotherapy is one of the most commonly used methods for treating cancer, but its side effects severely limit its application and impair treatment effectiveness. Removing off-target chemotherapy drugs from the serum promptly through adsorption is the most direct approach to minimize their side effects. In this study, we synthesized a series of adsorption materials to remove the chemotherapy drug doxorubicin by modifying MOF nanosheets with sulfonated azocalix[4]arenes. The strong affinity of sulfonated azocalix[4]arenes for doxorubicin results in high adsorption strength (Langmuir adsorption constant = 2.45–5.73 L mg^−1^) and more complete removal of the drug. The extensive external surface area of the 2D nanosheets facilitates the exposure of a large number of accessible adsorption sites, which capture DOX molecules without internal diffusion, leading to a high adsorption rate (pseudo-second-order rate constant = 0.0058–0.0065 g mg^−1^ min^−1^). These adsorbents perform effectively in physiological environments and exhibit low cytotoxicity and good hemocompatibility. These features make them suitable for removing doxorubicin from serum during “drug capture” procedures. The optimal adsorbent can remove 91% of the clinical concentration of doxorubicin within 5 min.

## 1. Introduction

Cancer has become one of the leading causes of death globally [1,2]. Chemotherapy remains one of the most common and powerful cancer treatments [3]. However, nearly all chemotherapy drugs have low selectivity. While inhibiting cancer cell growth, chemotherapy drugs are also toxic to healthy cells, which triggers various side effects [4,5,6,7]. For example, doxorubicin (DOX), which is widely used in the treatment of various hematological and solid tumor malignancies [8,9], could cause cardiomyopathy, congestive heart failure, and kidney damage [10,11,12,13,14]. The trade-off between therapy effectiveness and side effects must be considered in determining treatment regimens and dosages, ultimately impacting patient survival. Although various localized drug delivery approaches, such as transarterial chemoembolization (TACE) [15,16], have been developed, a significant portion of the drug can still pass through the tumor and flow through the body with the bloodstream. The implementation of the “drug capture” or “ChemoFilter” technique, which involves removing chemotherapy drugs before they enter systemic circulation, represents a direct strategy for addressing this issue [17,18]. Since the key to this technology lies in the adsorbent used, various adsorption materials targeting chemotherapy drugs have been developed [19,20,21,22,23]. For example, Grubbs and co-workers synthesized an adsorption material based on DNA and iron oxide particles that could remove 98% of DOX from human serum within 10 min, and the effectiveness of the material was proven in a porcine model [19]. Sheikhi et al. reported a cellulose-based nanoadsorbent for capturing DOX. This adsorbent could adsorb 6000 mg of DOX per 1 mg, demonstrating its high adsorption capacity [23].

Besides capacity, adsorption strength and rate are also important parameters for characterizing the performance of adsorbents. A higher adsorption strength leads to more thorough removal of the chemotherapy drug at equilibrium and less severe side effects. Introducing units with a strong binding affinity to the adsorbate is an effective strategy to enhance adsorption strength. It was reported that sulfonated azocalix[4]arene (SAC4A), a macrocyclic compound with a deep cavity, has a strong affinity for forming a host–guest complex with DOX (*K*_a_ = 6.47 × 10^6^ M^−1^) [24,25,26,27]. Thus, it offers a good option as a binding site for the development of DOX-targeted adsorbents. On the other hand, the adsorption rate determines the actual quantity of drug removed within the time it takes for the blood to pass through the adsorption device. The majority of current adsorbents are porous materials [28,29,30,31]. Adsorbate molecules must diffuse through the channels before coming into contact with the binding sites, which significantly impedes the adsorption kinetics. Using adsorbent materials with a large external surface area can expose a significant quantity of accessible adsorption sites on the surface, thereby addressing this issue. Two-dimensional metal–organic framework (MOF) nanosheets have unique advantages in this context. Their ultrathin sheet-like morphology maximizes the external surface area, and their flexible composition and modifiability make them easy to adjust according to the structure and properties of the targeted substance [32,33,34]. The potential of MOF nanosheets has been intensively investigated in recent years [35,36,37,38].

In this research, through the modification of the nanosheets of NUS-8, a stable Hf-based MOF with 2D connection, with SAC4A, a series of adsorption materials were prepared for the capture of DOX. Up to 86% of the adsorption capacities of these materials could be achieved within 5 min. Their Langmuir adsorption constants range from 2.45 to 5.73 L mg^−1^, indicating that their adsorption strength is higher than that of the reported adsorbents. One of these materials was screened and successfully utilized for removing DOX from an actual serum sample.

## 2. Materials and Methods

### 2.1. Materials and Apparatus

The chemicals used in the experiments, including hafnium chloride (HfCl_4_), 1,3,5-Tri(4-carboxyphenyl)benzene (H_3_BTB), 2′-amino-5′-(4-carboxyphenyl)-[1,1′:3′,1″-terphenyl]-4,4″-dicarboxylic acid (H_3_BTB-NH_2_), 1,3,5-tris(4,4,5,5-tetramethyl-1,3,2-dioxaborolan-2-yl)benzene, methyl 4-iodo-3-nitrobenzoate, palladium acetate (Pd(AcO)_2_), tetrabutylammonium bromide ((Bu)_4_NBr), tripotassium phosphate (K_3_PO_4_), iron, sodium bicarbonate (NaHCO_3_), lithium hydroxide hydrate (LiOH·H_2_O), 2-(7-aza-1H-benzotriazole-1-yl)-1,1,3,3-tetramethyluronium hexafluorophosphate (HATU), *N*,*N*-diisopropylethylamine (DIPEA), sodium carbonate (Na_2_CO_3_), calix[4]arene (C4A), 4-aminobenzenesulfonic acid, 4-aminobenzoic acid, sodium nitrite (NaNO_2_), sodium sulfate (Na_2_SO_4_), 4-aminobenzenesulfonic acid, sodium acetate trihydrate (CH_3_COONa·3H_2_O), hydrochloride (DOX), sodium chloride (NaCl), calcium chloride (CaCl_2_), bovine serum albumin (BSA), and Triton X-100, were purchased from Aladdin and used without further purification unless otherwise noted. 1,3,5-tris (2-amino-4-carboxyphenyl) benzenebenzene (H_3_BTB-3NH_2_), SAC4A, and monocarboxyl- and sulfonate-modified azocalix[4]arene (CSAC4A) were synthesized following literature procedures (Appendix A) [39,40,41]. H_3_BTB-3NH_2_ was synthesized through the Suzuki coupling reaction between 1,3,5-tris(4,4,5,5-tetramethyl-1,3,2-dioxaborolan-2-yl)benzene and 3-amino-4-bromobenzoic acid. SAC4A was prepared from the diazo-coupling reaction between C4A and 4-aminobenzenesulfonic acid. CSAC4A was synthesized using two steps of a diazo-coupling reaction, in which 4-aminobenzoic acid and 4-aminobenzenesulfonic acid were connected to C4A in sequence.

^1^H NMR measurements were carried out on a Bruker 400 MHz NMR spectrometer, and all chemical shifts are reported in parts per million (ppm). Powder X-ray diffraction (PXRD) patterns were recorded on a Rigaku D/max-2500 diffractometer with Cu K*α* radiation (λ = 0.15406 nm) at 40 kV and 100 mA. The 2*θ* range was from 2° to 50° with a 0.02° increment and a scan speed of 15°·min^−1^. Scanning electron microscopy (SEM) images were recorded on a TESCAN MIRA LMS scanning electron microscope. High-resolution transmission electron microscopy (HRTEM) data were collected using a FEI Talos F200S transmission electron microscope. Atomic force microscopy (AFM) images were observed by a Bruker Dimension Icon scanning electron microscope. Inductively coupled plasma optical emission spectrometry (ICP-OES) measurements were acquired on an Agilent 5110. N_2_ adsorption–desorption analysis was performed using a Micromeritics APSP 2460 surface area and porosity analyzer. The samples were degassed under vacuum at 120 °C for 12 h before measurement. Thermogravimetric analysis (TGA) was conducted using the Netzsch STA2500 Regulus and Netzsch STA449F3 Thermogravimetric Analyzer, with a heating rate of 10 °C/min from 30 °C to 80 °C under atmospheric air. Fourier transform infrared spectroscopy (FTIR) spectra were recorded on a Thermo Scientific Nicolet iS20 Spectrometer using KBr pellets dispersed with sample powders in the range of 4000–400 cm^−1^. X-ray photoelectron spectroscopy (XPS) data were measured on a Thermo ESCALAB 250XI equipped with an Al source anode X-ray gun. Absorption spectra (UV-vis) were collected under a Shimadzu UV-2450 spectrophotometer.

### 2.2. Synthesis of MOF Nanosheets

NUS-8 nanosheets were synthesized following a reported method with little modification [42]. HfCl_4_ (56 mg, 0.176 mmol) and 1,3,5-benzenetribenzoic acid (H_3_BTB) (50 mg, 0.114 mmol) were dissolved in a mixed solvent of DMF (8 mL), deionized water (1 mL), and formic acid (4 mL). The solution was placed in a 20 mL Pyrex vial, ultrasonicated for 5 min, and then heated at 120 °C for 48 h. After cooling to room temperature, the resulting white solid was centrifuged, washed with DMF (5 × 6 mL), solvent-exchanged with dichloromethane (5 × 6 mL) and methanol (5 × 6 mL) in sequence, and dried under vacuum at 120 °C for 24 h.

The synthesis of NUS-8-NH_2_ and NUS-8-3NH_2_ nanosheets followed a similar process to that of NUS-8, with the only difference being the replacement of the ligand by 2′-amino-5′-(4-carboxyphenyl)-[1,1′:3′,1″-terphenyl]-4,4″-dicarboxylic acid (H_3_BTB-NH_2_) and H_3_BTB-3NH_2_, respectively.

### 2.3. Modification of MOF Nanosheets by SAC4A/CSAC4A

The SAC4A/CSAC4A loaded adsorbents were synthesized through the post-synthesis modification of MOF nanosheets [43]. In the synthesis of SAC4A-loaded NUS-8 nanosheets (N@SAC4A), as-prepared NUS-8 nanosheets (50 mg) were dispersed in DMF (10 mL) by 30 min of ultrasonication. Then, a solution of SAC4A (50 mg) in DMF (10 mL) was added to the dispersion. The mixture was ultrasonicated for 3 h and then stirred at room temperature for 48 h. The resulting light orange solid product was collected, washed by DMF and CH_2_Cl_2_, solvent-exchanged with methanol for 24 h, and vacuum-dried at 100 °C for 24 h.

CSAC4A-linked NUS-8-NH_2_ (NA-CSAC4A) was synthesized by the following procedures: CSAC4A (50 mg) and *N*,*N*-diisopropylethylamine (DIPEA, 14.5 µL) were dissolved in 10 mL of DMF, stirred at room temperature for 30 min, and further stirred for 1 h after the addition of 2-(7-aza-1H-benzotriazole-1-yl) -1,1,3,3-tetramethyluronium hexafluorophosphate (HATU, 20.2 mg). The obtained solution was added to the suspension of NUS-8-NH_2_ (50 mg in 10 mL DMF). After 48 h of reaction, the light orange solid product formed was centrifuged, washed by DMF and CH_2_Cl_2_, solvent-exchanged with methanol for 24 h, and then vacuum-dried at 100 °C for 24 h.

The synthetic procedure for CSAC4A-linked NUS-8-3NH_2_ was similar to that of NA-CSAC4A, with the exception that the substrate was replaced by NUS-8-3NH_2_, and the addition amounts of CSAC4A, DIPEA, and HATU were changed to 100 mg, 29 µL, and 40.4 mg, respectively.

### 2.4. Adsorption Experiments

These experiments were performed with procedures similar to those described in [44,45]. A specific amount of adsorbent was immersed in a 5 mL aqueous solution of DOX and stirred for 24 h to achieve sorption equilibrium. The influence of adsorbent dosage on the adsorption process was investigated using 1 to 4 mg of adsorbent, while keeping the initial DOX concentration constant at 50 µM. Several solutions of DOX with concentrations ranging from 10 to 110 µM were mixed with 1.5 mg adsorbent to test the impact of the initial adsorbate concentration. To investigate the kinetics of the adsorption process, the experiment was carried out on a suspension with an initial volume of 50 mL, 15 mg of adsorbent, and 50 µM of DOX. At different time intervals, 3 mL of the suspension was sampled and analyzed.

The adsorption performance was also studied in the presence of various interfering substances. DOX solutions with pH values adjusted to 4–8 were used to assess the impact of solution pH. NaCl (80–190 mM) and CaCl_2_ (3–90 mM) were selected to mimic ion interference in the serum, while BSA (15–55 mg mL^−1^), the primary protein component in the blood, was tested as the protein interference (physiological levels of Na^+^, Ca^2+^, and BSA are approximately 140 mM, 3 mM, and 35 mg mL^−1^, respectively).

After the adsorption process, the adsorbent was separated by filtration. The residual concentration of DOX in the supernatant was quantified using UV-Vis spectroscopy at 480 nm. The removal efficiency (*R*) and adsorption amount (*q*_t_) were calculated by the following equations:(1)$$R(\%)=\frac{C_{0}-C_{t}}{C_{0}}\times 100\%$$
(2)$$q_{t}=\frac{(C_{0}-C_{t})\cdot V\cdot M}{m}$$
where *C*_0_ and *C*_t_ are the concentrations of DOX before and after the adsorption process, respectively; *V* is the volume of the DOX solution; *M* is the molecular weight of DOX; and *m* is the mass of the adsorbent.

### 2.5. Cytotoxicity Tests

As a reported method [46], the Cell Counting Kit-8 (CCK-8) assay was used to evaluate the cytotoxicity of N3A-CSAC4A. Human umbilical vein endothelial cells (HUVECs) were seeded into a 96-well plate at a density of 5000 cells per well and cultured in EGM-2 culture medium at 37 °C with 5% CO_2_ for 24 h. After the cells adhered completely, the culture medium was changed to a fresh medium with N3A-CSAC4A (0.1–1.2 mg/mL), and the cells were incubated for another 24 h. Following the addition of CCK-8 (20 μL) and a further 4 h incubation, the absorbance at 450 nm was recorded using a microplate reader.

### 2.6. Hemolysis Assay

As a reported method [46], the hemolysis assay was performed to evaluate the blood compatibility of N3A-CSAC4A. Fresh blood extracted from the abdominal aorta of mice was stored in an anticoagulant tube containing heparin. PBS was added at a volume about 10 times that of the blood, and the mixture was centrifuged at 3000 rpm for 5 min to wash the red blood cells (RBCs) three times. The RBCs were diluted to a concentration of 2% with PBS and then packed into microcentrifuge tubes. Then, the RBCs (200 μL) were incubated with 800 μL of N3A-CSAC4A (0.05–1.5 mg mL^−1^) at 37 °C for 4 h. We used PBS as the negative control and Triton X-100 (0.1%) as the positive control. After centrifugation, the supernatant (100 μL) was pipetted into a 96-well plate for absorbance testing at 545 nm using a microplate reader.

### 2.7. DOX Capture from Serum

As a typical procedure [47], the serum used in this experiment was extracted from the blood of mice, and the initial DOX solution was diluted to a clinical concentration of 86 μM by serum. N3A-CSAC4A (20 mg) was dispersed in 30 mL of the above DOX solution using 5 min of sonication and then stirred at room temperature. At different intervals, 3 mL of the suspension was sampled. After the adsorbent was separated, the absorbance at 480 nm was measured using UV-Vis spectroscopy.

## 3. Results and Discussion

### 3.1. Adsorbent Design

NUS-8, which is constructed by the metal nodes of Hf_6_ clusters and the ligand of H_3_BTB, was selected as the substrate for the adsorbents due to its superior stability compared to other reported 2D MOF nanosheets [48]. Two protocols were tried to load SAC4A onto NUS-8 nanosheets (Figure 1). The first one involves non-covalent binding. According to the hard–soft acid–base (HSAB) theory, the phenolic hydroxyl O atoms on the lower rim of SAC4A are hard bases and thus have a high affinity for the Hf_6_ clusters (hard acid) of NUS-8. To enhance the anchoring strength, we also attempted covalent bonding. To form an amide bond between SAC4A and MOF, one of the sulfo groups on the upper rim of the SAC4A molecule is replaced by a carboxyl group (the product is denoted as CSAC4A), and an amino group is prelocated on the H_3_BTB ligand. Two aminated ligands, H_3_BTB-NH_2_ and H_3_BTB-3NH_2_, with different substituent positions and numbers, were tested to screen for the adsorbent with the best performance.

### 3.2. Characterization

MOF nanosheets of NUS-8, NUS-8-NH_2_, and NUS-8-3NH_2_ were synthesized through the solvothermal reaction of HfCl_4_ with H_3_BTB, H_3_BTB-NH_2_ and H_3_BTB-3NH_2_, respectively. Powder X-ray diffractometry (PXRD) was first used to examine the crystallinity of the products. The recorded pattern closely matched the simulated pattern from the reported structure of NUS-8 [42,48], confirming its successful synthesis and high phase purity (Figure 1a). The PXRD patterns of NUS-8-NH_2_ and NUS-8-3NH_2_ are similar to those of NUS-8, indicating their isomorphic structures. Scanning electron microscopy (SEM) and transmission electron microscopy (TEM) images (Figure 1b and Appendix A) showed that these MOF samples present 2D layered morphologies. Hexagonal honeycomb patterns were observed in the high-resolution TEM (HR-TEM) images, confirming their long-range ordered crystalline structures. The pore size is approximately 2 nm, consistent with the backbone structures of these MOFs. The thicknesses of MOF nanosheets were measured using atomic force microscopy (AFM). The AFM images show the ultrathin structures with thicknesses of 1.38, 1.42, and 1.23 nm for NUS-8, NUS-8-NH_2_, and NUS-8-3NH_2_, respectively (Figure 1d). All three samples exhibited good chemical stability. The immersion of these samples in several common solvents for 48 h did not cause any significant changes in their PXRD patterns (Appendix A).

By loading SAC4A on MOF nanosheets through non-covalent and covalent interactions, three adsorbents were prepared. The PXRD patterns of N@SAC4A (SAC4A loaded NUS-8), NA-CSAC4A (CSAC4A-linked NUS-8-NH_2_), and N3A-CSAC4A (CSAC4A-linked NUS-8-3NH_2_) matched well with their precursors, indicating that the crystallinities of MOF nanosheets were not affected. The SEM images and TEM images revealed that the SAC4A/CSAC4A-modified products also exhibit lamellar morphologies (Figure 1b,c and Appendix A). All of these results confirm that the introduction of SAC4A/CSAC4A does not break the backbone of MOF precursors. The three modified samples exhibited absorption peaks at 1596, 1120, and 1035 cm^−1^ in the FT-IR spectra, which could be assigned to the stretching vibrations of N=N, S=O, and S−O, respectively, confirming the successful introduction of SAC4A/CSAC4A. In the spectrum of N@SAC4A, the intensity of the O−H stretching vibration around 3430 cm^−1^ was significantly reduced compared to that of NUS-8, suggesting the substitution of the OH^−^s on the Hf_6_ clusters by SAC4A. The vibrations of N−H and C=O at 1680 and 1275 cm^−1^ in the spectra of NA-CSAC4A and N3A-CSAC4A confirm the formation of an amide bond between CSAC4A and the aminated ligand. The –NH_2_ groups of precursors NUS-8-NH_2_ and NUS-8-3NH_2_ do not give prominent peaks in the spectra because their absorption is overlapped by that of OH^−^.

To further verify the introduction of SAC4A/CSAC4A, we conducted X-ray photoelectron spectroscopy (XPS) analysis (Appendix A). The S2p spectra of three SAC4A/CSAC4A-modified MOF samples are nearly identical, with two deconvoluted peaks at 167.4 and 168.5 eV attributed to the C−S and S=O bonds, respectively. This suggests that these samples contain the sulfur element from the macrocycle modifier. In the spectra of NUS-8-NH_2_ and NUS-8-3NH_2_, the N1s band can be resolved into two peaks at 400.2 and 401 eV, which are assigned to the C−N and N−H bonds, respectively. For NA-CSAC4A and N3A-CSAC4A, the N 1s spectrum displays three different chemical environments corresponding to N=N, C−N, and N−H bonds. The C−N/N−H area ratios of these two samples are higher than those of the corresponding precursors, suggesting the conversion of amino groups to amide bonds.

Inductively coupled plasma optical emission spectrometry (ICP-OES) was used to analyze the elemental compositions of SAC4A/CSAC4A-modified MOF samples. Since S and Hf are specific to the macrocycle modifier and MOF substrate, respectively, the loading amount of SAC4A/CSAC4A could be calculated from the ratio between these two elements. As shown in Appendix A, the CSAC4A loading amount of NA-CSAC4A (9.26 wt%) was obviously higher than the SAC4A loading amount of N@SAC4A (7.53 wt%), demonstrating that covalent bonding is a more effective method for the functionalization of MOFs. N3A-CSAC4A further increased the loading amount to 16.69 wt%, which indicates that the increase in anchoring sites is beneficial for the modification. We also altered the addition quantity of CSAC4A in the synthesis of N3A-CSAC4A and observed that it has a positive correlation with CSAC4A loading amount.

Upon the introduction of SAC4A/CSAC4A, the Brunauer–Emmett–Teller (BET) surface areas of NUS-8, NUS-8-NH_2_, and NUS-8-3NH_2_ decreased from 634, 440, and 259 m^2^ g^−1^ to 451, 398, and 241 m^2^ g^−1^, respectively. This reduction reveals that some of the modifier molecules enter into the pores of the nanosheets (Appendix A).

### 3.3. Adsorbent Studies

To test the removal efficiency for DOX, three MOF nanosheet precursors and their modified products with different dosages (ranging from 0.2 to 0.8 mg mL^−1^) were individually incubated in DOX solutions (50 µM) for 24 h. Each adsorbent decreased the concentration of DOX in a dose-dependent manner (Figure 2a). The adsorption percentages of three MOF precursors followed the order of NUS-8-NH_2_ < NUS-8-3NH_2_ < NUS-8. The highest adsorption capacity of NUS-8 could be explained by its large surface area (Appendix A). The superior performance of NUS-8-NH_2_ to NUS-8-3NH_2_ may be due to its less positive zeta potential, which weakens the electrostatic repulsion with the positively charged DOX (Appendix A). All three SAC4A/CSAC4A-modified MOF samples gave higher adsorption percentages than the corresponding precursors, which could be attributed to the strong complexation ability of the macrocycle introduced. It is expected that the DOX molecule enters the cavity of SAC4A/CSAC4A and forms a host–guest complex on the MOF surface. The increased amplitudes of adsorption amounts are in accordance with the SAC4A/CSAC4A loading amounts in these samples (Appendix A). With the addition of 0.3 mg mL^−1^ of N3A-CSAC4A, the adsorption percentage of DOX could reach 90%, which is 2.7 times higher than that of NUS-8-3NH_2_. The further increase in macrocycle loading through the enhancement of CSAC4A concentration in the synthesis system was proven to be unnecessary. When the modifier amount was doubled, the obtained product only gave a 3% increase in the adsorption percentage of DOX (Appendix A).

The DOX removal capabilities of these samples were further examined under different initial concentrations of DOX. As shown in Figure 2c, for all six adsorbents, the adsorbed amount of DOX per unit weight of adsorbent increased as the initial concentration changed from 10 to 70 µM. However, the removal percentage showed an opposite trend, which can be attributed to the depletion of adsorption sites (Appendix A). The adsorbed amount hardly changed when the DOX initial concentration was above 70 µM, indicating saturation had been reached.

To illustrate the thermodynamic information of the adsorption process, we tested the Langmuir (Equation (3)) and Freundlich (Equation (4)) adsorption isotherms to fit the equilibrium adsorption data:(3)$$q_{e}=\frac{K_{L}\cdot Q_{\max}\cdot C_{e}}{1+K_{L}\cdot C_{e}},$$
(4)$$q_{e}=K_{F}{C_{e}}^{1/n},$$

In these equations, *q*_e_ is the amount of DOX adsorbed at equilibrium, *Q*_max_ is the maximum adsorption capacity, *C*_e_ is the concentration of DOX at equilibrium, *K*_L_ is the Langmuir constant, *K*_F_ is the Freundlich constant, and *n* is the exponential factor of the Freundlich isotherm. The fitting curves of DOX adsorption are presented in Figure 2d, and the isotherm parameters of the two models are listed in Table 1.

For the three MOF nanosheets, the correlation coefficients (*R*^2^s) of the Freundlich isotherm model (0.9454–0.9732) are higher than the corresponding values of the Langmuir isotherm model (0.8271–0.9201), indicating that the Freundlich isotherm model is more suitable for describing the adsorption process. Their *Q*_max_s showed a positive correlation with the surface area, suggesting that DOX molecules are adsorbed onto the external surface of MOF nanosheets. In contrast to the cases of precursors, the Langmuir isotherm model of SAC4A/CSAC4A-modified products gave greater *R*^2^s (0.9636–0.9871) compared to the Freundlich isotherm model (0.7256–0.9605). The experimental maximum adsorption capacities were 70.5, 63.7, and 121.7 mg g^−1^ for N@SAC4A, NA-CSAC4A, and N3A-CSAC4A, respectively, which are close to the experimentally measured maximum adsorption capacities (69.3, 63.6, and 122.1 mg g^−1^; the slightly larger values of the measured adsorption capacities of N@SAC4A and NA-CSAC4A than the corresponding *Q*_max_s could be attributed to experimental error). These all indicate that the process of adsorption of DOX by SAC4A/CSAC4A-modified MOFs can be more accurately represented by the Langmuir isotherm model. Since the Langmuir isotherm model is generally associated with the homogeneity and the Freundlich isotherm is related to the heterogeneity of the adsorbent [49,50], the fitting results indicate that the macrocycle moieties offer uniform adsorption sites for DOX. The *Q*_max_s of SAC4A/CSAC4A-modified MOF nanosheets were generally higher than those of precursors. Among all the tested samples, N3A-CSAC4A gave the highest *Q*_max_ of 122.1 mg g^−1^, which is consistent with its highest macrocycle loading amount.

A more remarkable change upon the loading of SAC4A/CSAC4A is the enhancement of adsorption strength. The *K*_L_s of three macrocycle-modified MOF nanosheet adsorbents, which indicate their affinity towards the adsorbate, range from 2.45 to 5.73 L mg^−1^. Compared with the precursors, the *K*_L_s of two covalently modified samples showed an increase of an order of magnitude, while the non-covalent sample also increased by two times. Table 2 presents a comparison between the *K*_L_s of our samples and those reported in recent studies. Obviously, compared with reported adsorbents, our samples exhibit the highest affinity toward DOX. The highest affinity of our adsorbents would lead to the most complete removal of the drug after adsorption.

Then, the kinetics of the adsorption process was examined. As shown in Figure 3a, all three SAC4A/CSAC4A-modified MOF nanosheet adsorbents exhibited a sharp increase in adsorption quantity within the first 5 min, reaching 78–86% of the maximum value. This performance competes with the highest efficiency observed among reported adsorbents [23]. The fast adsorption process could be attributed to the ultrathin nanosheet morphology of these samples, which exposes the modified macrocycle moieties on the external surface to directly capture DOX. We used pseudo-first-order (Equation (5)), pseudo-second-order (Equation (6)), and intraparticle diffusion (Equation (7)) models to fit the temporal change in the adsorbed amount of DOX:(5)$$\ln(q_{e}-q_{t})=\ln q_{e}-k_{1}t,$$
(6)$$\frac{t}{q_{t}}=\frac{1}{k_{2}{q_{e}}^{2}}+\frac{1}{q_{e}}t,$$
(7)$$q_{t}=k_{\mathrm{id}}t^{0.5}+C$$
where *q*_e_ and *q*_t_ are the quantities of DOX adsorbed at equilibrium and time *t*; *k*_1_, *k*_2_, and *k*_id_ represent the rate constants of pseudo-first-order, pseudo-second-order, and intraparticle diffusion equations, respectively; and *C* is the thickness constant of the boundary layer.

The fitting graphs of the three models are presented in Figure 3b–d, and the corresponding parameters are listed in Table 3. The *R*^2^s of the pseudo-second-order model for all three modified adsorbents reach 0.99, and the fitted values of *q_e_*s from this model closely align with the experimental results. These all demonstrate that the pseudo-second-order model provides the best fit for the adsorption process of DOX. Typically, pseudo-second-order adsorption kinetics suggest a chemisorption mechanism. When the DOX molecules reached the surface of modified MOF nanosheets, they promptly bonded with the SAC4A/CSAC4A moieties through host–guest complexation. The uniform environment of the SAC4A/CSAC4A moiety, along with its high affinity and 1:1 stoichiometry in forming a guest–host complex with DOX, enables the capture of DOX, demonstrating the feature of chemosorption.

### 3.4. Application to the DOX Capture from Serum

Because N3A-CSAC4A showed the best adsorption performance in the above experiments, the following investigations focused on this adsorbent.

The normal pH value of human serum ranges from 7.35 to 7.45, and various substances, such as ions and proteins, are present in serum. To evaluate the potential of N3A-CSAC4A in drug capture, a series of experiments was conducted to examine its adsorption performance under interference conditions. N3A-CSAC4A was first exposed to a series of DOX solutions whose pH values were adjusted to 4–8 (Figure 4a). It was observed that the adsorption percentage of DOX increased as the pH of the solution changed from 4 to 7 and then remained stable at around 90%, indicating that N3A-CSAC4A can achieve the optimum effect at normal serum pH. The reduction in adsorption percentage at lower pH values could be explained by the electrostatic interaction between the adsorbent (its zeta potential changed from −39.1 to 14.4 mV as the pH decreased from 8 to 4, Appendix A) and the positively charged DOX. Then, the adsorption efficiency of N3A-CSAC4A was tested in the presence of Na^+^, Ca^2+^, and BSA (Figure 4b and Appendix A). The results showed that under normal concentrations of Na^+^ and Ca^2+^ in serum, the adsorption efficiency is close to that of the blank control. With the increase in BSA concentration, the adsorption percentage even showed a slight rise. These results prove that the complex components of serum cannot impair the adsorption performance of N3A-CSAC4A.

In chemotherapy drug capture, the adsorbent is immobilized and comes into direct contact with blood, which imposes stringent requirements for its cytotoxicity and hemocompatibility. To assess the cytotoxicity of N3A-CSAC4A, various concentrations of its suspensions were incubated with human umbilical vein endothelial cells (HUVECs, Figure 4c). Even though the concentration of N3A-CSAC4A increased to 1.2 mg mL^−1^, the cell viability could still be maintained above 90% after 24 h of incubation. This result indicates that the cytotoxicity of N3A-CSAC4A is negligible. In the hemolytic study, the red blood cells were incubated with varying concentrations of N3A-CSAC4A, and the hemolysis ratio was measured with Triton X-100 as a positive control (Figure 4d). N3A-CSAC4A at a concentration of 1.5 mg mL^−1^ only gave a hemolysis ratio of 1.59 ± 0.75%, indicating that this adsorbent is non-toxic to erythrocytes.

Encouraged by the good adsorption performance of N3A-CSAC4A in a physiological environment, as well as its good biocompatibility and hemocompatibility, we used it to capture DOX from real serum. DOX was added to serum to a clinical concentration of 86 μM, and its concentration change was monitored after the addition of 0.67 mg mL^−1^ N3A-CSAC4A (Figure 5). A rapid DOX removal process was observed in the initial 5 min, with the adsorption percentage surging to 91%. After this period, the concentration decline of DOX slowed down, with the majority of DOX (97%) being adsorbed within 60 min. These results demonstrate that N3A-CSAC4A could be used as an efficient and rapid adsorbent for the removal of DOX from serum.

## 4. Conclusions

In summary, we developed a series of adsorbents by modifying MOF nanosheets with SAC4A/CSAC4A. The incorporated SAC4A/CSAC4A moieties serve as strong binding sites for DOX, and the extensive external surface of the MOF nanosheet substrate facilitates contact with the adsorbate, thereby expediting the adsorption kinetics. Our experiments demonstrated that, compared with non-covalent modification, covalent bonding is a more efficient approach for functionalizing MOF nanosheets, giving products with increased stability, a higher modifier loading amount, and enhanced adsorption capacity. The developed adsorbent exhibited favorable attributes such as strong anti-interference capability, low cytotoxicity, and good hemocompatibility, enabling its successful application in DOX capture from real serum samples. The results of this study suggest that, by utilizing abundant data on host–guest recognition of macrocycles, the functionalization of MOF nanosheets with macrocycles is an adaptable strategy for fabricating targeted adsorbents.

## Data Availability

Data are contained within the article.

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
