# Peer review of "Sulfonated Azocalix[4]arene-Modified Metal–Organic Framework Nanosheets for Doxorubicin Removal from Serum"

_nanomaterials, 2024, doi:10.3390/nano14100864_

Round 1

Reviewer 1 Report

Comments and Suggestions for Authors

The manuscript named "Sulfonated Azocalix[4]arene Modified Metal-Organic Framework Nanosheets for Doxorubicin Removal from Serum" presents original and interesting results. However, the authors failed to include two references, 8 and 9, in the Introduction. It would be helpful to list all reagents used in the synthesis, including HfCl4, under the Materials and Apparatus section.

Author Response

To reviewer 1:

Remarks:

The manuscript named "Sulfonated Azocalix[4]arene Modified Metal-Organic Framework Nanosheets for Doxorubicin Removal from Serum" presents original and interesting results. However, the authors failed to include two references, 8 and 9, in the Introduction. It would be helpful to list all reagents used in the synthesis, including HfCl4, under the Materials and apparatus section.

Response:

Thank reviewer 1 for his/her comments on the manuscript. Ref. 8 and 9 are cited in the revised introduction (…which is widely used in the treatment of various hematological and solid tumor malignancies [8,9],…). All the reagents used in the experiments are listed in the Materials and Apparatus section of the revised manuscript:

“The chemicals used in the experiments, including hafnium chloride (HfCl4), 1,3,5-Tri(4-carboxyphenyl)benzene (H3BTB), 2'-amino-5'-(4-carboxyphenyl)-[1,1':3',1''-terphenyl]-4,4''-dicarboxylic acid (H3BTB-NH2), 1,3,5-tris(4,4,5,5-tetramethyl-1,3,2-dioxaborolan-2-yl)benzene, methyl 4-iodo-3-nitrobenzoate, palladium acetate (Pd(AcO)2), tetrabutylammonium bromide ((Bu)4NBr), tripotassium phosphate (K3PO4), iron, sodium bicarbonate (NaHCO3), lithium hydroxide hydrate (LiOH·H2O), 2-(7-aza-1H-benzotriazole-1-yl)-1,1,3,3-tetramethyluronium hexafluorophosphate (HATU), ethyldiisopropylamine (DIPEA), sodium carbonate (Na2CO3), calix[4]arene (C4A), 4-aminobenzenesulfonic acid, 4-aminobenzoic acid, sodium nitrite (NaNO2), sodium sulfate (Na2SO4), 4-aminobenzenesulfonic acid, sodium acetate trihydrate (CH3COONa·3H2O), hydrochloride (DOX), sodium chloride (NaCl), calcium chloride (CaCl2), bovine serum albumin (BSA), and Triton X-100, were purchased from Aladdin and used without further purification unless otherwise noted. 1,3,5-tris (2-amino-4-carboxyphenyl) benzenebenzene (H3BTB-3NH2), SAC4A, and monocarboxyl- and sulfonate-modified azocalix[4]arene (CSAC4A) were synthesized following literature procedures (Figures. S1−S3) [39−41].”

Reviewer 2 Report

Comments and Suggestions for Authors

The paper entitled “Sulfonated Azocalix[4]arene Modified Metal-Organic Frame-2 work Nanosheets for Doxorubicin Removal from Serum” was reviewed. The research contains preparation of new adsorbents based on nanosheets with or without macrocyclic compounds. The authors found new and useful findings for removal of doxorubicin as drug for tumor. The paper can be published from Nanomaterials, but there are still unclear points. So it can be published from Nanomaterials after adequate revision.

1.     The most important point is how calixarene was connected with nanosheets and how doxorubicin was caught by such materials. Doxorubicin molecule is too large to be caught by the cavity of calix[4]arene. So, what is the contribution of calixarene scaffold for uptake of doxorubicin?

2.     How about drug toxicity of nanomaterials themselves?

3.     In the last sentence of Introduction, NUS-8 was introduced. But, at this point, the authors did not define NUS-8.

4.     Please add abbreviations of all nanomaterials in Scheme 1.

5.     Related to comment 1, NUS-8-3NH2 and NUS-8-3NH2 in Figure 1(e) do not have two peaks around 3500 cm-1 as typical primary amino peak.

6.     Does the adsorbed amount of doxorubicin relate to the introduced amount of calixarene or amino groups?

7.     The authors discussed adsorption model by using Freundlich and Langmuir models. Langmuir is for the adsorption using mono-layer. How about Freundlich? And what does mean that Freundlich model is preferable?

Comments on the Quality of English Language

I have no idea to give comments, because I am not native speaker. But the sentences are easily understood.

Author Response

To reviewer 2:

Remarks:

The paper entitled “Sulfonated Azocalix[4]arene Modified Metal-Organic Framework Nanosheets for Doxorubicin Removal from Serum” was reviewed. The research contains preparation of new adsorbents based on nanosheets with or without macrocyclic compounds. The authors found new and useful findings for removal of doxorubicin as drug for tumor. The paper can be published from Nanomaterials, but there are still unclear points. So it can be published from Nanomaterials after adequate revision.

Response:

Thank reviewer 2 for his/her comments on the manuscript.

Remarks:

1. The most important point is how calixarene was connected with nanosheets and how doxorubicin was caught by such materials. Doxorubicin molecule is too large to be caught by the cavity of calix[4]arene. So, what is the contribution of calixarene scaffold for uptake of doxorubicin?

Response:

In this research, sulfonated azocalix[4]arene (SAC4A), instead of calix[4]arene, was used is the modifier. With the extension of –N=N−(C6H4)− structure, the depth of the cavity of calix[4]arene is increased to 12.2 Å (Figure R1), which is comparable to the molecular size of DOX. As a result, SAC4A exhibits a high affinity towards DOX. In our previous works, we have measured the equilibrium constant for the formation of a host-guest complex between SAC4A and DOX, which is as high as 6.47 × 106 M-1 (Mater. Horiz. 2023, 10, 1689−1696). With this high affinity, we believe that the loaded CSAC4A/SAC4A units could function as strong binding sites for DOX.

For the sample of N@SAC4A, SAC4A is connected to MOF nanosheets through the coordination of the hydroxyl groups with the Hf6 clusters. The substitution of the OH−s on the Hf6 clusters by SAC4A is evidenced by the reduced intensity of the O−H stretching vibration (3430 cm−1) in the FT-IR spectrum of N@SAC4A relative to that of the precursor. For NA-CSAC4A and N3A-CSAC4A, CSAC4A is connected to MOF nanosheets by an amide bond, which forms between the pre-located amino group on the MOF and the carboxyl group of CSAC4A. The formation of an amide bond is indicated by the increased C−N/N−H area ratio in the XPS spectrum and the presence of peaks corresponding to N−H (1680 cm−1) and C=O (1275 cm−1) in the FT-IR spectra.

In the revised manuscript, the deep cavity feature of SAC4A is highlighted and the binding constant between SAC4A and DOX is provided to support the high affinity of SAC4A towards this drug.

Figure R1. Structure of SAC4A, optimized at B3LYP/6-31G(d) level.

Remarks:

2. How about drug toxicity of nanomaterials themselves?

Response:

To evaluate the toxicity of N3A-CSAC4A, its cytotoxicity and hemocompatibility was tested. As shown in Fig. 4c and 4d, the cell viability maintained above 90% after 24 h of incubation with 1.2 mg mL−1 N3A-CSAC4A, and 1.5 mg mL−1 of this adsorbent can only give a hemolysis ratio of 1.59 ± 0.75 %. These results shows the good biocompatibility and hemocompatibility of N3A-CSAC4A. Since N3A-CSAC4A is insoluble, and the adsorbent is immobilized during drug capture (meaning the adsorbent only comes into contact with blood during treatment), I believe these experiments are sufficient to demonstrate the safety of N3A-CSAC4A.

In the revised manuscript, the rationale for evaluating drug toxicity using cytotoxicity and hemocompatibility assays is explained.

Remarks:

3. In the last sentence of Introduction, NUS-8 was introduced. But, at this point, the authors did not define NUS-8.

Response:

According to the suggestion of the reviewer, a brief introduction about NUS-8 is added here:

“In this research, through the modification of the nanosheets of NUS-8, a stable Hf-based MOF with 2D connection, with SAC4A, a series of adsorption materials was prepared for the capture of DOX.”

Remarks:

4. Please add abbreviations of all nanomaterials in Scheme 1.

Response:

The abbreviations of all nanomaterials have been supplemented to Scheme 1.

Remarks:

5. Related to comment 1, NUS-8-3NH2 and NUS-8-3NH2 in Figure 1(e) do not have two peaks around 3500 cm-1 as typical primary amino peak.

Response:

Thank the reviewer for pointing out this problem. Beside amino groups, these samples also have a plethora of OH ions on the Hf6 clusters. OH ions produce a broad absorption band across the range of 3200−3600 cm-1, which overlaps with the absorption positions of the amino group. As a result, the amino groups in these samples do not give prominent peaks in the spectra.

In the revised manuscript, the reason for the absence of the absorption peaks of amino group is explained.

Remarks:

6. Does the adsorbed amount of doxorubicin relate to the introduced amount of calixarene or amino groups?

Response:

As shown in Figure S10, the adsorbed amount of DOX generally has a positive correlation with the loading amount of calixarene modifier. This result is highlighted in the revised manuscript to further prove that the calixarene modifier functions as the binding sites for DOX.

Remarks:

7. The authors discussed adsorption model by using Freundlich and Langmuir models. Langmuir is for the adsorption using mono-layer. How about Freundlich? And what does mean that Freundlich model is preferable?

Response:

It generally believed that Langmuir isotherm model is related to the homogeneity and Freundlich isotherm is related to the heterogeneity of the adsorbents (J. Environ. Chem. Eng 7 (2019) 103130 and J. Appl. Polym. Sci. 114 (2009) 2139–2148). Therefore, if the Freundlich isotherm model were more suitable for describing the adsorption process, this fitting result would indicate that the adsorbent could provide binding sites with varying affinities for the adsorbate. The meaning of Freundlich model has been added to the revised manuscript:

“Since the Langmuir isotherm model is generally associated with the homogeneity and the Freundlich isotherm is related to the heterogeneity of the adsorbent, the fitting results…”

Reviewer 3 Report

Comments and Suggestions for Authors

I carefully reviewed this manuscript. This manuscript has interesting results and findings on adsorptive removal of doxorubicin. However, this manuscript is not completed. There are many revisions before publications. Unfortunately, I can’t accept it like this because my comments for revisions are too many. I think English quality is favorable for publication. Therefore, the composition of the manuscript and analysis of experimental results and data are required tobe modified or rearranged.

Abstract

Authors should explain the abstract of this manuscript with more numerical values, or experimental data in order to increase the understanding for readers.

L36-38: References 8 and 9 are not cited.

L27 and 49: the terms of “minutes” and “min” as unit are used. It should be unified.

L47: Reference 19 is not written by Grubbs et al. The author names are different.

The reference that has the first name of Grubbs is not shown.

L50: Reference 23 is not written by Sheikhi et al. The author names are different. There is not a reference by Sheikhi et al.

The authors must carefully review the all references.

L80: In general, the manufacturer(s) for the used chemicals is required to be shown in the section 2.1.

L83: “Fig. S1-S3” is changed to “Figs. S1-S3”.

L81-84: Short descriptions for the procedure for synthesis of these compounds are required for increasing the reader’s understanding.

L85: This sentence is not correct. To be correct, “1H NMR measurements were carried out”.

L123: “was dispersed in DMF” is better.

L108-139: On the synthesis of these compounds, some references are required. It is not common to synthesize these compounds without any reference.

Section 2.2-2.7: for a series of the procedures and experiments, references are not cited. I think that the author(s) should have referred to some references.

L192: Reference [42] is not cited in the body text.

L209: “pattern closely matched” Evidence is required, for example, use a reference, show the result(s) in a figure in the body text or supplementary material.

L213, 219, and 222: The terms “Figure” and “Fig.” are mixed. It should be unified.

L222: Fig. S4 is not shown and not explained.

L231: Here is Fig. S4. The numerical is rearranged in order. The terms “Figure” and “Fig.” are mixed. It should be unified.

L243: “S 2p” is changed to “S2p” or “S2p”.

L248: “N 1s” is changed to “N1s” or “N1s”.

L192, 253, 274, 338,371: The use of the first person “we” is not favorable.

L269: The scale of the x and y axis starts from 0 (zero). In Figure 2 (a), (c), and (d), the same symbol and color should be used for the same sample.

L307: The unit of LF is not correct.

L317-319: In this case, the maximum adsorption capacity is fixed to the experimental adsorption capacity. This means that all adsorption cites were used for DOX adsorption. In general, the maximum adsorption capacity is higher than the experimental adsorption capacity.

L361: The scale of the x and y axis starts from 0 (zero).

L365: The unit of C is not shown. Was the kid value calculated from which part(s) of the curve(s) shown in Figure 3(d). The x axis in Figure 3(d) must be shown in t0.5 (min0.5).

Section 3.2 and 3.3: Can the author(s) show or explain a possible adsorption mechanism?

L 424: Figures S1-S12 are shown or explained in the body text. However, Figures S13-S16 are not explained and not cited in the body text.

Comments on the Quality of English Language

Unfortunately, I can’t accept it like this because my comments for revisions are too many. I think English quality is favorable for publication. Therefore, the composition of the manuscript and analysis of experimental results and data are required to be modified or rearranged.

Author Response

To reviewer 3:

Remarks:

I carefully reviewed this manuscript. This manuscript has interesting results and findings on adsorptive removal of doxorubicin. However, this manuscript is not completed. There are many revisions before publications. Unfortunately, I can’t accept it like this because my comments for revisions are too many. I think English quality is favorable for publication. Therefore, the composition of the manuscript and analysis of experimental results and data are required tobe modified or rearranged.”

Response:

Thank reviewer 3 for his/her comments on the manuscript.

Remarks:

1. Abstract

Authors should explain the abstract of this manuscript with more numerical values, or experimental data in order to increase the understanding for readers.

Response:

According to the suggestion of the reviewer, more experimental data have been added to the abstract of revised manuscript:

“…The strong affinity of sulfonated azocalix[4]arenes for doxorubicin results in high adsorption strength (Langmuir adsorption constant = 2.45−5.73 L mg−1) and more complete removal of the drug. The extensive external surface area of the 2D nanosheets facilitates the exposure of a large quantity of accessible adsorption sites, which capture DOX molecules without internal diffusion, leading to a high adsorption rate (pseudo-second-order rate constant = 0.0058−0.0065 g mg−1 min−1).”

Remarks:

2. L36-38: References 8 and 9 are not cited.

Response:

References. 8 and 9 are cited in the revised introduction (…which is widely used in the treatment of various hematological and solid tumor malignancies [8,9],…).

Remarks:

3. L27 and 49: the terms of “minutes” and “min” as unit are used. It should be unified.

Response:

These terms have been unified to “min” in the revised manuscript.

Remarks:

4. L47: Reference 19 is not written by Grubbs et al. The author names are different.

The reference that has the first name of Grubbs is not shown.

Response:

We have re-checked this reference and confirmed that Robert H. Grubbs is listed as the corresponding author of this reference.

Remarks:

5. L50: Reference 23 is not written by Sheikhi et al. The author names are different. There is not a reference by Sheikhi et al. The authors must carefully review the all references.

Response:

We have re-checked this reference. Ref. 23 is a paper published by Sarah A.E. Young, Joy Muthami, Mica Pitcher, Petar Antovski, Patricia Wamea, Robert Denis Murphy, Reihaneh Haghniaz, Andrew Schmidt, Samuel Clark, Ali Khademhosseini and Amir Sheikhi, and Amir Sheikhi is the corresponding author of this manuscript.

Remarks:

6. L80: In general, the manufacturer(s) for the used chemicals is required to be shown in the section 2.1.

Response:

The sources of the used chemicals have been supplemented to the revised manuscript:

“The chemicals used in the experiments, including hafnium chloride (HfCl4), 1,3,5-Tri(4-carboxyphenyl)benzene (H3BTB), 2'-amino-5'-(4-carboxyphenyl)-[1,1':3',1''-terphenyl]-4,4''-dicarboxylic acid (H3BTB-NH2), 1,3,5-tris(4,4,5,5-tetramethyl-1,3,2-dioxaborolan-2-yl)benzene, methyl 4-iodo-3-nitrobenzoate, palladium acetate (Pd(AcO)2), tetrabutylammonium bromide ((Bu)4NBr), tripotassium phosphate (K3PO4), iron, sodium bicarbonate (NaHCO3), lithium hydroxide hydrate (LiOH·H2O), 2-(7-aza-1H-benzotriazole-1-yl)-1,1,3,3-tetramethyluronium hexafluorophosphate (HATU), ethyldiisopropylamine (DIPEA), sodium carbonate (Na2CO3), calix[4]arene (C4A), 4-aminobenzenesulfonic acid, 4-aminobenzoic acid, sodium nitrite (NaNO2), sodium sulfate (Na2SO4), 4-aminobenzenesulfonic acid, sodium acetate trihydrate (CH3COONa·3H2O), hydrochloride (DOX), sodium chloride (NaCl), calcium chloride (CaCl2), bovine serum albumin (BSA), and Triton X-100, were purchased from Aladdin and used without further purification unless otherwise noted.”

Remarks:

7. L83: “Fig. S1-S3” is changed to “Figs. S1-S3”.

Response:

This error has been corrected accordingly.

Remarks:

8. L81-84: Short descriptions for the procedure for synthesis of these compounds are required for increasing the reader’s understanding.

Response:

We have added the short descriptions for the synthesis procedures of H3BTB-3NH2, SAC4A and CSAC4A:

“H3BTB-3NH2 was synthesized through the Suzuki coupling reaction between 1,3,5-tris(4,4,5,5-tetramethyl-1,3,2-dioxaborolan-2-yl)benzene and 3-amino-4-bromobenzoic acid. SAC4A was prepared from the diazo-coupling reaction between C4A and 4-aminobenzenesulfonic acid. CSAC4A was synthesized using two steps of diazo-coupling reaction, in which 4-aminobenzoic acid and 4-aminobenzenesulfonic acid were connected to C4A in sequence.

Remarks:

9. L85: This sentence is not correct. To be correct, “ 1H NMR measurements were carried out”.

Response:

This sentence has been corrected accordingly.

Remarks:

10. L123: “was dispersed in DMF” is better.

Response:

This sentence has been revised accordingly.

Remarks:

11. L108-139: On the synthesis of these compounds, some references are required. It is not common to synthesize these compounds without any reference. Section 2.2-2.7: for a series of the procedures and experiments, references are not cited. I think that the author(s) should have referred to some references.

Response:

The synthesis of MOF precursors (section 2.2) consulted Angew. Chem. Int. Ed. 2016, 55, 4962−4966. The modification of MOF nanosheets by SAC4A/CSAC4A (section 2.3) used the methods summarized in J. Mater. Chem. A 2023, 11, 24519−24550. Adsorption experiments were conducted with the procedures similar to those in Chem. Eng. J. 2018, 353, 482−489 and Environ. Sci. Pollut. Res. 2022, 29, 35012−35024. The cytotoxicity tests (section 2.5) and hemolysis assay (section 2.6) followed the similar procedures to those in Int. J. Biol. Macromol. 2023, 236, 123942. The removal of DOX from serum was performed with the same method as that in ACS Omega 2020, 5, 29121−291264. Above mentioned literatures are cited in the revised manuscript.

Remarks:

12. L192: Reference [42] is not cited in the body text.

Response:

Reference [42] is cited in L130 of the revised manuscript.

Remarks:

13. L209: “pattern closely matched” Evidence is required, for example, use a reference, show the result(s) in a figure in the body text or supplementary material.

Response:

References are cited here to support this discussion:

“The recorded pattern closely matched the simulated pattern from the reported structure of NUS-8 [42,48],…”

….

“[42] Cao, L.; Lin, Z.; Peng, F.; Wang, W.; Huang, R.; Wang, C.; Yan, J.; Liang, J.; Zhang, Z.; Zhang, T.; Long, L.; Sun, J.; Lin, W. Self-supporting metal-organic layers as single-site solid catalysts. Angew. Chem. Int. Ed. 2016, 55, 4962−4966.

[48] Hu, Z.; Mahdi, E. M.; Peng, Y.; Qian, Y.; Zhang, B.; Yan, N.; Yuan, D.; Tan, J.-C.; Zhao, D. Kinetically controlled synthesis of two-dimensional Zr/Hf metal-organic framework nanosheets via a modulated hydrothermal approach. J. Mater. Chem. A 2017, 5, 8954−8963.”

Remarks:

14. L213, 219, and 222: The terms “Figure” and “Fig.” are mixed. It should be unified.

Response:

These terms have been unified to “Figure” in the revised manuscript.

Remarks:

15. L222: Fig. S4 is not shown and not explained.

  1. L231: Here is Fig. S4. The numerical is rearranged in order. The terms “Figure” and “Fig.” are mixed. It should be unified.

Response:

We have re-checked the order of all the figures in the manuscript. The terms “Figure” and “Fig.” are unified to “Figure” in the revised manuscript.

Remarks:

17. L243: “S 2p” is changed to “S2p” or “S2p”.

Response:

We have changed “S 2p” to “S2p” in the revised manuscript (L268).

Remarks:

18. L248: “N 1s” is changed to “N1s” or “N1s”.

Response:

We have changed “N 1s” to “N1s” in the revised manuscript (L272).

Remarks:

19. L192, 253, 274, 338,371: The use of the first person “we” is not favorable.

Response:

These sentences have been revised: “We tried two protocols to load…” has been changed to “Two protocols were tried to load…” (L219), “We used inductively coupled plasma optical emission spectrometry (ICP-OES)…” has been changed to “Inductively coupled plasma optical emission spectrometry (ICP-OES) was used…” (L278), “…we individually incubated three MOF nanosheet precursors and their modified products with different dosages (ranging from 0.2 to 0.8 mg mL−1)…” has been changed to “…three MOF nanosheet precursors and their modified products with different dosages (ranging from 0.2 to 0.8 mg mL−1) were individually incubated…” (L294), “…we examine the kinetics of the adsorption process.” has been changed to “…the kinetics of the adsorption process was examined.” (L366), and “…we conducted a series of experiments…” has been changed to “…a series of experiments was conducted…” (L402).

Remarks:

20. L269: The scale of the x and y axis starts from 0 (zero). In Figure 2 (a), (c), and (d), the same symbol and color should be used for the same sample.

Response:

The scales of the x and y axes in Figures 2a, 2c and 2d have been adjusted to start from zero. The symbol and color in these figures are also changed according to the suggestion of the reviewer (Figure R2).

Figure R2. Revised Figure 2.

Remarks:

21. L307: The unit of LF is not correct.

Response:

The unit of KF has been corrected to mg L1/n μmol−1/n g–1.

Remarks:

22. L317-319: In this case, the maximum adsorption capacity is fixed to the experimental adsorption capacity. This means that all adsorption cites were used for DOX adsorption. In general, the maximum adsorption capacity is higher than the experimental adsorption capacity.

Response:

The slightly larger values of the measured adsorption capacities of N@SAC4A and NA-CSAC4A (70.5 and 63.7 mg g−1) than the corresponding Qmaxs of these adsorbents (69.3 and 63.6 mg g−1) could be attributed to experimental error. Because the high affinity of these adsorbents toward DOX (5.73 and 2.86 L μmol−1) and the high adsorbate concentration tested in our experiments (110 μM), the adsorption capacity is expected to be very close to Qmax. Considering the experimental error, the slightly higher value of measured adsorption capacity is possible. This abnormal phenomenon is explained in the revised manuscript:

“…the slightly larger values of the measured adsorption capacities of N@SAC4A and NA-CSAC4A than the corresponding Qmaxs could be attributed to experimental error…”

Remarks:

23. L361: The scale of the x and y axis starts from 0 (zero).

Response:

The scales of the x and y axes in Figures 3a, 3c and 3d have been adjusted to start from zero (Figure R3). For Figure 3b, because several data points have negative y values, only the scale of the x axis of this figure is changed.

Figure R3. Revised Figure 3.

Remarks:

24. L365: The unit of C is not shown. Was the kid value calculated from which part(s) of the curve(s) shown in Figure 3(d). The x axis in Figure 3(d) must be shown in t 0.5 (min 0.5 ).

Response:

According to Eq. 7, C has the same unit as that of qt (mg g−1). The unit of C has been supplemented to the revised manuscript. The kid value was obtained from the fitting of the data points in the whole tested time range (0−300 min). Figure 3d has been re-drawn with the x axis of t0.5 (min0.5) (Figure R3)

Remarks:

  1. Section 3.2 and 3.3: Can the author(s) show or explain a possible adsorption mechanism?

Response:

A possible adsorption mechanism has been added:

“All three SAC4A/CSAC4A-modified MOF samples gave higher adsorption percentages than the corresponding precursors, which could be attributed to the strong complexation ability of the macrocycle introduced. It is expected that the DOX molecule enters the cavity of SAC4A/CSAC4A and forms a host-guest complex on the MOF surface.”

Remarks:

26. L424: Figures S1-S12 are shown or explained in the body text. However, Figures S13-S16 are not explained and not cited in the body text.

Response:

In the revised manuscript, Figures S13−S16 are cited in the maintext:

“When modifier amount was doubled, the obtained product only gave a 3% increase of the adsorption percentage of DOX (Figures S12−S14).

The reduction in adsorption percentage at lower pH values could be explained by the electrostatic interaction between the adsorbent (its zeta potential changed from −39.1 to 14.4 mV as the pH decreased from 8 to 4, Figure S15) and the positively charged DOX.

Then, the adsorption efficiency of N3A-CSAC4A was tested in the presence of Na+, Ca2+, and BSA (Figures 4b and S16).”

Round 2

Reviewer 2 Report

Comments and Suggestions for Authors

The revised paper entitled “Sulfonated Azocalix[4]arene Modified Metal-Organic Frame-2 work Nanosheets for Doxorubicin Removal from Serum” was re-reviewed. The authors well answered and revised the manuscript well. Now, the paper can be published from Nanomaterials as it is.

Reviewer 3 Report

Comments and Suggestions for Authors

The authors properly revised the manuscript.

L82-104: Some large (long) white spaces should be shortened or become short by using hyphenation. I agree to publish this manuscript in this journal.